# Recombinant FGF21 Attenuates Polychlorinated Biphenyl-Induced NAFLD/NASH by Modulating Hepatic Lipocalin-2 Expression

**DOI:** 10.3390/ijms23168899

**Published:** 2022-08-10

**Authors:** Hye Young Kim, Young Hyun Yoo

**Affiliations:** Department of Anatomy and Cell Biology and BK21 Program, Department of Translational Biomedical Science, Dong-A University College of Medicine, Busan 49201, Korea

**Keywords:** FGF21, hepatic iron overload, lipocalin-2, NAFLD, NASH, PCBs, TASH

## Abstract

Although recent studies have demonstrated that polychlorinated biphenyls (PCB) exposure leads to toxicant-associated steatohepatitis, the underlying mechanism of this condition remains unsolved. Male C57Bl/6 mice fed a standard diet (SD) or 60% high fat diet (HFD) were exposed to the nondioxin-like PCB mixture Aroclor1260 or dioxin-like PCB congener PCB126 by intraperitoneal injection for a total of four times for six weeks. We observed hepatic injury, steatosis, inflammation, and fibrosis in not only the Aroclor1260-treated mice fed a HFD but the PCB126-treated mice fed either a SD or a HFD. We also observed that both types of PCB exposure induced hepatic iron overload (HIO). Noticeably, the expression of hepatic lipocalin-2 (LCN2) was significantly increased in the PCB-induced nonalcoholic fatty liver disease (NAFLD)/nonalcoholic steatohepatitis (NASH) models. The knockdown of LCN2 resulted in improvement of PCB-induced lipid and iron accumulation in vitro, suggesting that LCN2 plays a pivotal role in PCB-induced NAFLD/NASH. We observed that recombinant FGF21 improved hepatic steatosis and HIO in the PCB-induced NAFLD/NASH models. Importantly, recombinant FGF21 reduced the PCB-induced overexpression of hepatic LCN2 in vivo and in vitro. Our findings indicate that recombinant FGF21 attenuates PCB-induced NAFLD/NASH by modulating hepatic lipocalin-2 expression. Our data suggest that hepatic LCN2 might represent a suitable therapeutic target for improving PCB-induced NAFLD/NASH accompanying HIO.

## 1. Introduction

Nonalcoholic fatty liver disease (NAFLD) is developing as a fast-growing health challenge, with a prevalence approaching 25% in the general population. The spectrum of NAFLD comprises multiplex abnormalities, ranging from a simple hepatic steatosis (NAFLD), to nonalcoholic steatohepatitis (NASH) which is characterized by hepatocellular injury, lobular inflammation, and apoptosis that can lead to fibrosis and cirrhosis [1,2]. While NAFLD is considered a benign and reversible stage arising from the excessive lipid accumulation in hepatocytes, NASH is a more progressive and severe stage of NAFLD and is irreversible. NAFLD/NASH is associated with an increased risk of hepatocellular carcinoma (HCC), cardiovascular diseases, and complications related to type 2 diabetes mellitus (T2DM), such as nephropathy and neuropathy [1,2,3]. Moreover, no pharmacological treatment has been approved for NAFLD. Thus, delineating the mechanisms explaining the pathogenesis of NAFLD is crucial for managing NAFLD/NASH and its comorbidities. NAFLD cannot be fully explained by overnutrition, genetics, pharmacology, or level of physical activity. Current studies have demonstrated additional contributing factors.

Polychlorinated biphenyls (PCBs), which constitute a class of persistent organic pollutants (POPs), have been demonstrated to play a role in the development of NAFLD/NASH [4,5]. The term toxicant-associated steatohepatitis (TASH) has been coined to describe steatohepatitis associated with chemical pollutants such as PCBs [6]. PCB congeners have been subclassified into the following two major categories based on their structure and elicited responses: dioxin-like (DL) and nondioxin-like (NDL) PCBs. DL PCBs have a coplanar structure and bind and activate the aryl hydrocarbon receptor (AhR) [7,8]. PCB 126 is the most potent and environmentally relevant DL PCB. PCB126 acts as an endocrine disruptor and has recently been reported to be associated with hepatic fat accumulation and inflammation [9,10]. NDL PCBs are noncoplanar and activate other xenobiotic receptors, such as the pregnane X receptor (PXR) and constitutive androstane receptor (CAR) [11]. The commercially manufactured PCB mixture Aroclor1260 is the best mimic of human PCB bioaccumulation patterns in human fat [12,13]. Previous studies have shown that Aroclor 1260 acts as a “second hit” in the conversion of diet-induced hepatic steatosis, leading to more advanced steatohepatitis in a chronic exposure mouse model [13]. Since low-dose Aroclor1260 does not activate either human or murine AhR, Aroclor1260 has previously been used to model NDL PCB exposure [7,11,13].

PCBs have also been classified as endocrine, metabolic, and signaling disrupting chemicals (EDCs/MDCs/SDCs) that contaminate the food supply and accumulate in human adipose tissue and liver [14]. EDCs interfere with any aspect of hormone action, while MDCs promote metabolic changes that can result in obesity, diabetes, fatty liver, or metabolic syndrome [15,16]. Indeed, multiple epidemiological studies have reported an association between PCB exposure in the body and elevated NAFLD biomarkers [4,5]. Several animal studies have suggested that PCBs cause NAFLD [11,13,17]. Despite these recent studies, the mechanisms underlying the pathogenesis of NAFLD by PCB exposure have not been fully delineated.

Iron is an essential nutrient with several biological functions, such as oxygen transport, mitochondrial function, and DNA synthesis. Recent studies have reported that the dysregulation of iron homeostasis may be associated with more advanced stages or a higher incidence of NAFLD/NASH [18,19,20]. Excess iron increases reactive oxygen species (ROS) production and cytotoxicity through lipid peroxidation, protein denaturation, and DNA damage [21]. Since the liver increases iron storage and protects other organs from iron-induced damage under conditions of excess iron, the liver is a major target organ for iron toxicity. Excess iron in the liver is associated with an increased severity and progression of liver diseases, including steatosis, steatohepatitis, fibrosis, cirrhosis and HCC, in patients with NAFLD. Hyperferritinemia associated with iron accumulation is frequently observed in NAFLD patients [22,23]. In several studies, hepatic iron overload (HIO) has been observed in patients with NAFLD and NASH [19,24,25]. We also previously observed HIO and hyperferritinemia in HFD-induced NAFLD models [26].

Lipocalin-2 (LCN2), also known as neutrophil gelatinase-associated lipocalin (NGAL), is a secreted 25-kDa glycoprotein that belongs to the lipocalin superfamily first identified as a protein stored in specific granules of human neutrophils and is an innate immune protein [27,28]. LCN2 belongs to a group of transporters of small lipophilic molecules, such as steroids, lipopolysaccharides, iron, and fatty acids, in circulation and has recently emerged as a new player in the regulation of host responses to inflammation, particularly in modulating iron homeostasis [28]. Accumulating evidence suggests that the altered expression of LCN2 plays critical roles in several pathological organ conditions, including liver injury and steatosis, renal damage, brain injury, cardiomyopathies, musculoskeletal disorders, lung infection, and cancer in several organs [28,29,30]. Recent studies have shown elevated LCN2 levels in serum, liver, and adipose tissue in *db*/*db* or high-fat diet–fed mice, obese individuals, and NASH patients [28,31]. LCN2 is significantly upregulated in several benign and malignant liver diseases, making it a good candidate for a NAFLD biomarker or even a therapeutic target [29].

In a previous study, we demonstrated that recombinant fibroblast growth factor 21 (FGF21), which is currently in clinical development for the potential treatment of NASH [32], ameliorates HFD-induced hepatic steatosis by improving HIO [26]. FGF21 is preferentially expressed in organs associated with metabolic functions, such as the liver, skeletal muscle, pancreas and adipose tissue. Recent studies have shown that the administration of an FGF21 analog to humans with obesity significantly lowered body weight and decreased LDL cholesterol and triglycerides while increasing HDL cholesterol [33]. Moreover, both recombinant FGF21 and FGF21 analogs have emerged as promising therapeutic drug candidates for metabolic disorders, including hepatic steatosis, obesity, insulin resistance and dyslipidemia, in several animal studies [34,35]. However, the mechanism by which recombinant FGF21 exerts a therapeutic effect on TASH is not yet entirely understood.

Here, we show that both DL- and NDL-PCBs induce hepatic steatosis, fibrosis, inflammation and HIO by upregulating the expression of hepatic LCN2, resulting in the induction of NAFLD/NASH in vivo and in vitro. We further demonstrate that recombinant FGF21 ameliorates hepatic lipid and iron accumulation in this NAFLD/NASH model by downregulating the PCB-induced overexpression of hepatic LCN2.

## 2. Results

### 2.1. PCB Exposure Induces NAFLD/NASH in Mice Fed a SD or HFD

#### 2.1.1. Aroclor1260 Exposure Induces NAFLD/NASH in Mice Fed a HFD

We first examined the hepatic effects of the NDL PCB mixture Aroclor1260, whose composition mimics human bioaccumulation patterns, in a mouse model of HFD-induced NAFLD/NASH. Male C57BL/6 mice were fed either a SD or 60% HFD for four weeks and then treated with vehicle (corn oil) or Aroclor1260 (20 mg/kg) by intraperitoneal injection for a total of four injections (two, three, four, and five weeks) during the six-week study duration (Figure 1A). Aroclor1260 exposure significantly induced various manifestations of hepatic steatosis (Figure 1B–F), including liver volume and weight gain (Appendix A), increased vacuolization/lipid accumulation (Figure 1B,C), hepatic TG (Figure 1D), hepatic FFA (Figure 1E) and elevated plasma levels of native FGF21 (Figure 1F), which is a potential plasma marker for diagnosing NAFLD, in the mice fed a HFD but not those fed a SD. Exposure to 20 μM Aroclor1260 also increased lipid accumulation in O/P-treated human ARE primary hepatocytes (Figure 1G). Recent studies suggest that adipocyte death and adipose tissue inflammation play an important role in triggering liver injury and inflammation and NAFLD/NASH progression [36,37]. In the present study, we observed that the Aroclor1260 exposure increased the number of dead adipocytes, which were detected as crown-like structures (CLS) and are markers of chronic inflammation in adipose tissues (Figure 1H,I), but not the adipocyte size (Figure 1J). Furthermore, the Aroclor1260 exposure induced fibrosis in the mice fed a HFD but not those fed a SD (Appendix A).

#### 2.1.2. PCB126 Exposure Induces NAFLD/NASH in Mice Fed a SD or HFD

Next, we examined the hepatic effects of the representative ND PCB congener PCB126 in a mouse model of HFD-induced NAFLD/NASH. Male C57BL/6 mice were fed either a SD or 60% HFD for four weeks and then treated with vehicle (corn oil) or PCB 126 (5 mg/kg) by intraperitoneal injection for a total of four injections (two, three, four, and five weeks) during the six-week study duration (Figure 1A). The PCB126 exposure induced various manifestations of hepatic steatosis (Figure 1B–F), including liver volume and weight gain (Appendix A), increased vacuolization/lipid accumulation (Figure 1B,C), hepatic TG (Figure 1D), hepatic FFA (Figure 1E) and elevated plasma levels of native FGF21 (Figure 1F) in mice fed a SD or HFD. Exposure to 10 μM PCB126 also increased lipid accumulation in O/P-treated human ARE primary hepatocytes (Figure 1G). The PCB126 exposure also increased the number of CLSs (Figure 1H,I) but not the adipocyte size (Figure 1J). In contrast to Aroclor1260, the PCB126 exposure induces NAFLD/NASH in the mice fed a SD. Furthermore, PCB126 exposure induced fibrosis in the mice fed a SD or HFD (Appendix A).

### 2.2. PCB Exposure Induces Hepatic Iron Overload

Subsequently, we observed iron accumulation in the liver tissue samples using Prussian blue staining (Figure 2A) and iron concentration measurements (Figure 2B) in livers obtained from Aroclor1260- and PCB126-exposed mice fed either a SD or HFD. These results indicate that PCB exposure induces HIO in mice fed either a SD or HFD. We further observed that treatment with 200 μM FAC, an iron source, augmented lipid accumulation in HepG2 cells cotreated with O/P (250 μM) (Figure 2C). Furthermore, ORO staining indicated that FAC treatment increased lipid accumulation in human ARE primary hepatocytes treated with either Aroclor1260 or PCB126 (Figure 2D). BODIPY 493/503 staining also showed that FAC treatment increased lipid accumulation in HepG2 cells treated with either Aroclor1260 or PCB126 (Figure 2E).

### 2.3. PCB Exposure Accelerates Hepatic Inflammation

Since NASH is liver inflammation and damage caused by a buildup of fat in the liver, we further examined the effect of PCB exposure on hepatic inflammation. We observed that both Aroclor1260 and PCB126 significantly increased the hepatic expression level of both mRNA (Figure 3A) and protein (Figure 3B) of mouse F4/80 in mice fed either a SD or HFD. Recently, it has been reported that DL and NDL PCBs differentially regulate the hepatic proteome and modify diet-induced NAFLD severity [38]. To dissect the molecular mechanisms mediating PCB-induced hepatic inflammatory effects in the injured liver, we analyzed the levels of 111 hepatic inflammatory cytokines using a mouse XL cytokine array (Figure 3C). Importantly, among the numerous hepatic proinflammatory cytokines significantly increased by Aroclor1260 in mice fed a HFD or PCB126 in mice fed either a SD or a HFD, the expression levels of lipocalin-2/NGAL, myeloperoxidase (MPO) and matrix metalloproteinase-9 (MMP-9) were commonly increased (Figure 3D). These findings suggest that at least one of these upregulated factors may play a role in the progression of hepatic steatosis to steatohepatitis in mice exposed to PCBs.

### 2.4. Recombinant FGF21 Improved Hepatic Steatosis in PCB-Induced NAFLD/NASH Models

Subsequently, we investigated the effect of recombinant FGF21 on hepatic steatosis in the PCB-induced NAFLD/NASH models. We observed that the administration of rmFGF21 (1 mg/kg/day) for 10 days (Appendix A) decreased serum ALT (Appendix A) and serum AST (Appendix A) and attenuated various manifestations of hepatic steatosis, including liver weight gain (Figure 4A–D) and increased vacuolization/lipid accumulation (Figure 4E). We further examined whether recombinant FGF21 regulates PCB-induced lipid accumulation in primary hepatocytes. To mimic in vivo NAFLD models accompanying HIO, FAC was administered to human ARE primary hepatocytes cotreated with O/P. Importantly, rhFGF21 (50 ng/mL, 24 h) significantly reduced FAC-induced lipid accumulation in PCB-exposed human ARE primary hepatocytes (Figure 4F). These results indicate that recombinant FGF21 improves hepatic steatosis in PCB-induced NAFLD/NASH models.

### 2.5. Recombinant FGF21 Attenuated HIO in PCB-Induced NAFLD/NASH Models

We observed the effect of recombinant FGF21 on HIO in PCB-induced NAFLD/NASH models. The administration of rmFGF21 (1 mg/kg/day) for 10 days significantly improved HIO in both types of PCB-induced NAFLD/NASH mouse models (Figure 5A,B). We also observed that rhFGF21 (50 ng/mL, 24 h) significantly attenuated the FAC-augmented hepatic iron accumulation in PCB-exposed HepG2 cells (Figure 5C). These results demonstrate that recombinant FGF21 attenuates HIO in PCB-induced NAFLD/NASH models.

### 2.6. Hepatic LCN2 Mediates PCB-Induced Hepatic Lipid and Iron Accumulation

Previous studies have demonstrated that LCN2 is a critical iron regulatory protein under physiological and inflammatory conditions [28]. Among the three upregulated inflammatory cytokines (Figure 3D), MPO and MMP-9 were not upregulated at the mRNA level. Thus, further assays focused on LCN2. We observed that the mRNA and protein expression levels of LCN2 were markedly increased in the liver tissues from the Aroclor1260- or PCB126-injected mice fed either a SD or HFD (Figure 6A,B). Additionally, the protein expression level of LCN2 was increased by both 20 μM Aroclor1260 and 10 μM PCB126 in both BSA- and O/P-treated HepG2 cells (Figure 6C). To explore the potential role of hepatic LCN2 in the development of PCB-induced NAFLD/NASH, we examined the effect of the in vitro knockdown of hepatic LCN2 using a siRNA delivery system. LCN2 deficiency attenuated PCB-induced hepatic lipid and iron accumulation in HepG2 cells (Figure 6D,E). These results suggest that hepatic LCN2 mediates PCB-induced hepatic inflammation and hepatic lipid and iron accumulation, suggesting that LCN2 is a key pathogenic regulator in PCB-induced NAFLD/NASH models.

### 2.7. Recombinant FGF21 Reduces PCB-Induced Overexpression of Hepatic LCN2

Subsequently, we explored whether recombinant FGF21 regulates the PCB-induced overexpression of hepatic LCN2. We observed that the administration of rmFGF21 (1 mg/kg/day) for 10 days significantly decreased the mRNA (Figure 7A) and protein (Figure 7B) expression of hepatic LCN2 in both Aroclor1260- and PCB126-injected mice fed either a SD or HFD. Notably, the treatment with rhFGF21 (50 ng/mL, 24 h) decreased the protein expression of LCN2 in 20 μM Aroclor1260- or 10 μM PCB126-treated HepG2 cells (Figure 7C). These results demonstrate that recombinant FGF21 reduces the PCB-induced overexpression of hepatic LCN2. Overall, these data indicate that recombinant FGF21 attenuates PCB-induced NAFLD/NASH by modulating hepatic LCN2 expression.

## 3. Discussion

The global epidemic of metabolic syndrome and NAFLD/NASH has largely been attributed to endogenous factors, such as genetics, diet, lifestyles and aging. However, there is also increasing evidence of the contributing role of environmental exposure [39,40] The coined term TASH [6] has emerged to reflect that a form of NAFLD/NASH caused in workers highly exposed to toxic pollutants is not rare. Exposure to environmental contaminants, including PCBs, bromodichloromethane (BDCM) and organochlorine pesticides, such as dieldrin and DDT, may play a role in the genesis and development of NAFLD or may act as a ‘second hit’ in the progression of hepatic steatosis to steatohepatitis [41,42,43]. Multiple human cohort studies have demonstrated associations between PCB exposure and suspected NAFLD [4,5,41], and animal studies have suggested the causal role of PCBs in NAFLD [10,13]. However, environmental hepatology remains an understudied field. The mechanisms by which chemical pollution impacts TASH pathogenesis are largely unknown.

PCBs are environmental pollutants that are manufactured and used commercially as dielectric fluids in transformers [14]. While the production of PCBs was banned in 1979, they still persist in the environment, including food, and continue to cause health problems. PCBs bioaccumulate in the liver and adipose tissue due to their hydrophobicity, thus rendering these sites principal targets of PCB toxicity. Recent epidemiologic studies have shown that PCB exposure can also result in metabolic disorders associated with NAFLD, including obesity, insulin resistance/diabetes, and metabolic syndrome [4,11,44]. Aroclor1260 is a PCB mixture that contains 60% chlorine by weight and was one of the Aroclors marketed in the early stages of PCB production. It was subsequently replaced by other Aroclors (1254, 1248 and 1242). Although several PCB mixtures have been commercially produced and widely used, we selected Aroclor1260 for this study due to the similarity in its congener composition pattern to that in human fat. We additionally selected a representative DL PCB, PCB126, since Aroclor1260 has previously been used to model NDL PCB exposures. A recent study demonstrated that both DL and NDL PCBs have been associated with NAFLD, but PCBs differentially regulate the hepatic proteome and modify diet-induced NAFLD severity [38], and their effects and mechanisms also differ. DL PCBs altered the gut-liver axis and microbiome and caused hepatic steatosis by disrupting hepatic lipid metabolism. Unexpectedly, NDL PCBs reduced the liver’s protective responses to promote diet-induced NAFLD [43].

In the present study, we used exposure to either the NDL PCB mixture Aroclor1260 or DL PCB126 congener in C57BL/6 mice. We observed that both the Aroclor1260-treated mice fed a HFD and PCB126-treated mice fed either a SD or a HFD showed similar phenotypes, such as liver damage, hepatic steatosis, fibrosis and inflammation, which are representative NAFLD/NASH manifestations. We also observed that both types of PCBs induced hepatic iron accumulation in vivo and in vitro. In addition, the treatment with an iron source, FAC, augmented the PCB-induced hepatic lipid accumulation in human ARE primary hepatocytes and HepG2 cells treated with O/P, indicating that excess iron increases lipid accumulation in hepatocytes. Our data obtained from the present study aiming to identify the common mechanism underlying two types of PCB-induced NAFLD/NASH suggest that dysregulated iron metabolism may play a pivotal role in the pathogenesis of PCB-induced NAFLD/NASH, indicating that the modulation of iron regulation could be a potential therapeutic strategy.

Recent studies have shown that iron metabolism is involved in the development of insulin resistance, dyslipidemia, and NAFLD/NASH. Some reports have demonstrated that patients with NAFLD/NASH have an increased likelihood of developing HIO with disrupted iron homeostasis [24,45,46] and that iron reduction therapies, such as phlebotomy and iron chelation, improve insulin resistance and liver function in patients with NAFLD [47,48]. Additionally, clinical evidence indicates that the response to infection and inflammation worsens with elevated iron stores [49]. Recent studies have demonstrated that exposure to environmental pollutants can lead to a disruption of the hepcidin–ferroportin axis along with disordered systemic iron homeostasis and diseases [50,51]. The data obtained support that both types of PCBs induced hepatic iron accumulation. We hypothesized that factors, such as cytokines causing inflammation in the pathogenesis of PCB-induced TASH, may be directly or indirectly involved in HIO. Although exposure to PCBs could lead to dysregulated inflammatory responses according to previous animal and human studies [52,53], systematic and quantitative analyses of the associations between different types of PCBs and markers of inflammation and immune responses have not been documented before the present study. Using a mouse cytokine array evaluating 111 cytokines/chemokines, we found that exposure to both types of PCBs commonly increased the expression levels of hepatic LCN2, MPO, and MMP-9 in vivo.

LCN2 and MPO are neutrophil-derived proteins. Neutrophil infiltration is very common in both patients and mice with NASH. Although how neutrophils contribute to the development of NAFLD is unclear, several reports have shown that the release of neutrophil-specific components is required for the progression of NAFLD [54]. For example, an important neutrophil-derived enzyme, MPO, was found to be increased in NASH patients. The accumulation of MPO and MPO-mediated oxidation products may contribute to liver inflammation and the development of NASH [55]. MPO is a heme-containing enzyme that uses hydrogen peroxide (H_2_O_2_) and halide ions to generate hypohalous acid, a potent oxidizing and antimicrobial agent. MPO is located within neutrophils with the siderophore-binding innate immune protein LCN2 [56]. MMP-9 plays an important role in extracellular matrix remodeling during hepatic fibrosis. MMP-9 also binds LCN2 and forms the MMP-9/LCN2 complex, preventing the degradation of MMP-9 [57]. These two cytokines, MPO and MMP-9, may act in concert with the LCN2 protein and are involved in liver diseases, such as NAFLD/NASH.

We analyzed the expression levels of LCN2, MPO, and MMP-9 in our PCB-induced NAFLD/NASH models. We observed that the mRNA and protein expression levels of hepatic LCN2 were increased by two types of PCBs (negative data of MPO and MMP-9 mRNA are not shown). A recent study showed that hepatic LCN2 is markedly increased in experimental liver injury and that its increase is mediated by increased liver injury after CCl_4_, ConA, and LPS exposure [58]. LCN2 has emerged as a critical iron regulatory protein under physiological and inflammatory conditions [28,59]. Some studies reported contradictory data. LCN2 deficiency protects against alcohol- and diet-induced liver inflammation and injury in mouse models of NASH [60,61], and LCN2-deficient mice are refractory to obesity-induced insulin resistance, adipose tissue inflammation and endothelial dysfunction [62]. In the present study, the knockdown of hepatic NCN2 using a siRNA delivery system resulted in improvement in PCB-induced hepatic steatosis and HIO in vitro models, indicating that LCN2 plays a critical role in PCB-induced NAFLD/NASH. Despite its role in iron homeostasis, LCN2 does not directly bind iron but interacts with iron only by forming a ternary complex with a siderophore as its cofactor [28]. Previously, studies focused on microbial siderophores, and the existence of mammalian siderophores has not been delineated to date.

Recombinant FGF21 and FGF21 analogs have emerged as promising therapeutic drug candidates for metabolic disorders, including NAFLD [63,64], and are currently in clinical development for the potential treatment of NASH [32]. In mice, recombinant FGF21 leads to a reduction in body weight and adiposity without a reduction in food intake. In addition, FGF21 agonists improve hepatic and systemic insulin sensitivity, leading to a reduction in hepatic steatosis [26,65]. In preclinical studies, FGF21 administration reduced hepatic steatosis and inflammation in different NASH models [66]. In the present study, we evaluated the therapeutic effect of recombinant FGF21 on PCB-induced NAFLD/NASH. We demonstrated that FGF21 ameliorates PCB-induced lipid and iron accumulation. Furthermore, we demonstrated that recombinant FGF21 reduced the PCB-induced overexpression of hepatic LCN2 in both in vivo and in vitro NAFLD/NASH models. Our data indicate that recombinant FGF21 attenuates polychlorinated biphenyl-induced NAFLD/NASH by modulating hepatic LCN2 expression.

Humans are exposed to very low-dose, complex mixtures of POPs. However, in the present study, we adopted the POP treatment for a short duration. Thus, whether the data obtained from our experimental system are relevant to human exposure is unclear. Despite these limitations, the present study provides mechanistic insight into the therapeutic effect of FGF21 on PCB-induced NAFLD/NASH. Our data also suggest that hepatic LCN2 might represent a suitable therapeutic target for improvement in PCB-induced NAFLD/NASH accompanying HIO. Further in vivo, in vitro and clinical studies are necessary to unveil the multiple roles of LCN2 in steatosis and iron dysregulation during inflammation.

In conclusion, recombinant FGF21 attenuates PCB-induced NAFLD/NASH by modulating hepatic LCN2 expression, and hepatic LCN2 may represent a suitable therapeutic intervention target for PCB exposure-induced NAFLD/NASH. Further studies should evaluate whether drugs interfering with LCN2 synthesis and/or biological actions have therapeutic effects in various TASH models.

## 4. Materials and Methods

### 4.1. Reagents and Antibodies

3,3′4,4′,5-pentachlorobiphenyl (PCB126) and PCB mixture, Aroclor1260 were purchased from AccuStandard Inc. (New Haven, CT, USA). Recombinant human FGF21 (rhFGF21) was purchased from Peprotech (Rocky Hill, NJ, USA), and recombinant murine FGF21 (rmFGF21) was purchased from Cusabio (Houston, TX, USA). Oleic acid (OA), palmitic acid (PA), free fatty acid (FFA)-free bovine serum albumin (BSA), Oil red O (ORO) and ferric ammonium citrate (FAC) were purchased from Sigma (St. Louis, MO, USA). TRIzol reagents and Lipofectamine^®^ RNAiMAX transfection reagent were purchased from Invitrogen (Carlsbad, CA, USA). BODIPY 493/503 was purchased from Molecular Probes (Eugene, OR, USA). FerroFarRed was purchased from GORYO Chemical (Hokkaido, Japan). The antibodies against Lipocalin-2/NGAL was purchased from BOSTER Biological Technology (Pleasanton, CA, USA), and antibodies against F4/80 was obtained from Thermo Fisher Scientific (Cleveland, OH, USA). Antibodies against GAPDH was purchased from Santa Cruz Biotechnology (Santa Cruz, CA, USA).

### 4.2. Animals

Male C57BL/6 mice were purchased from SamTako Bio-Korea (Osan, Korea). The animals were maintained in a temperature-controlled room (22 °C) on a 12:12 h light–dark cycle. Eight-week-old C57Bl/6 mice were fed a standard diet (SD) for a total of 4 kcal per 1 g of diet or a 60% high-fat diet (HFD) for a total of 5.33 kcal per 1 g of diet. The HFD comprised 20% carbohydrate (0% sucrose), 20% protein, and 60% fat. Twelve-week-old mice were treated with vehicle (corn oil), Aroclor 1260 (20 mg/kg) or PCB 126 (5 mg/kg) by intraperitoneal (ip) injection for a total of four injections (2, 3, 4 and 5 weeks) to ensure that acute toxicity was minimized during the 6-week study duration. In the National Toxicology Program (NTP) studies, a 20 mg/kg cumulative dose yielded the following tissue levels: serum—176 ng/g, liver—3663 ng/g and adipose—92,840 ng/g. Thus, the 20 mg/kg dose employed is expected to produce serum levels similar to the maximum levels reported in the Anniston cohort (170.4 ng/g). We considered 20 mg/kg the maximum concentration, and through preliminary experiments, we determined the concentration that induces NAFLD while minimizing toxicity [67]. The mice were randomly divided into several groups of 10 animals each. The mice were intraperitoneally injected with vehicle or rmFGF21 (1 mg/kg/day) once daily for 10 days as follows: (1) SD or HFD treatment with an equivalent volume of saline administered intraperitoneally (SD/HFD-vehicle); (2) SD or HFD treatment with Aroclor1260 administered intraperitoneally (SD/HFD-Aroclor1260); (3) SD or HFD treatment with PCB126 administered intraperitoneally (SD/HFD-PCB126); (4) SD or HFD treatment with rmFGF21 and saline administered intraperitoneally (SD/HFD-rmFGF21); (5) SD or HFD treatment with Aroclor1260 and rmFGF21 administered intraperitoneally (SD/HFD-Aroclor1260-rmFGF21); and (6) SD or HFD treatment with PCB126 and rmFGF21 administered intraperitoneally (SD-PCB126-rmFGF21). After six weeks of treatment, the mice were sacrificed after overnight fasting, and serum was collected from the retro-orbital sinus of each non-anesthetized mouse. All HFD components were purchased from FeedLab (Guri, Korea). All procedures were approved by the Committee on Animal Investigations at Dong-A University (DIACUC-15-12).

### 4.3. Cell Culture and Treatment of Free Fatty Acids

Assay-Ready Expanded (ARE) human primary hepatocytes were purchased from Axol Bioscience (Cambridge, UK). Primary cultures were obtained by hepatocyte seeding at a density of 3 × 10^4^ cells/cm^2^ onto collagen I-precoated plates in maintenance medium plus supplement (Axol Bioscience) for 24 h at 37 °C. The medium was then replaced the maintenance medium, and the cells were cultured for three days before proceeding with endpoint assays. HepG2 cells were obtained from the American Type Culture Collection (Manassas, VA, USA) and were cultured as previously described [68]. Each fatty acid was dissolved in ethanol and diluted in Dulbecco’s modified Eagle’s medium containing 0.1% (*w*/*v*) fatty-acid-free BSA. In addition, a mixture of the fatty acids (O/P) containing oleic acid (66%) and palmitic acid (33%) was prepared. Fatty acid–BSA complexes was added to the serum-containing cell culture medium to achieve a final concentration of 250 μM. The controls were incubated with equal concentrations of FFA-free BSA containing ethanol. To examine the effect of several chemicals, the cells were co-treated with 250 μM O/P for 24 h.

### 4.4. RNA Interference and Transfection

ON-TARGETplus human LCN2 siRNA (NM_005564) was purchased from Dharmacon (Thermo Scientific, Hudson, NH, USA). As a negative control, the same nucleotides were scrambled to form nongenomic combinations. HepG2 cells were transfected with siRNAs using Lipofectamine^®^ RNAiMAX transfection reagent (Invitrogen) according to the manufacturer’s instructions.

### 4.5. Histological Analysis

Liver and adipose tissues were fixed in 10% neutral buffered formalin and embedded in paraffin. Histological and immunohistochemical staining for mouse tissue samples were performed as previously described [68]. Staining for iron content in liver sections was carried out by using a Perl’s Prussian blue staining kit (Polysciences, Inc., Warrington, PA, USA). Staining for collagen in liver sections was carried out by using a Picrosirius Red staining kit (Polysciences). The histological images were observed and analysed using a Pannoramic MIDI-II Digital Scanner (3D Histech, Budapest, Hungary). The quantification of lipid droplets in the liver and adipose tissues sections were measured by the Image J software version 1.53s (National Institute of Health, Bethesda, MD, USA), which utilizes the color and shape of the H&E staining images to determine the percentage of lipid droplets in the whole field.

### 4.6. Analysis of Metabolites

Plasma FGF21 was measured using a mouse FGF21 ELISA kit (MyBioSource, San Diego, CA, USA). Hepatic FFA and TG levels were measured using a quantification assay kit (Abcam, Cambridge, MA, USA). The hepatic iron concentrations of mice were measured at 50 mg liver tissue using an iron colorimetric assay kit (BioVision, Milpitas, CA, USA). Hepatic cytokine/chemokine profiling was performed at 150 μg liver tissue using Proteome Profiler mouse XL cytokine array (R&D system, Minneapolis, MN, USA).

### 4.7. Fluorescence Staining and Confocal Microscopy

For the staining of lipid droplets, HepG2 cells were cultured and treated on a coverslip and incubated with diluted BODIPY 493/503 for 30 min at 37 °C. For the staining of Fe^2+^, HepG2 cells were cultured and treated on a coverslip and incubated with diluted FerroFarRed for 1 h at 37 °C. Fluorescence images were observed and analyzed under a Zeiss LSM 700 laser-scanning confocal microscope (Gottingen, Germany).

### 4.8. Oil Red O Staining

Cells were washed twice in PBS and fixed for 1 h using 4% (*w*/*v*) paraformaldehyde at room temperature. After three washes in 60% isopropanol, the cells were stained for 20 min with freshly diluted ORO solution. Then, the stain was removed, and the cells were washed four times in distilled water. To quantify intracellular lipid accumulation, ORO solution was extracted using 100% isopropanol, and the absorbance of the eluted ORO was measured at 500 nm using a spectrophotometer.

### 4.9. Western Blot Analysis

Cells were washed twice with ice-cold PBS, resuspended in ice-cold RIPA buffer and incubated at 4 °C for 30 min. Lysates were centrifuged at 13,000 rpm for 20 min at 4 °C. Equal amounts of proteins were subjected to 7.5–15% sodium dodecyl sulfate polyacrylamide gel electrophoresis. The proteins were transferred to a nitrocellulose membrane (Amersham Pharmacia Biotech, Piscataway, NJ, USA) and reacted with each antibody. Immunostaining with antibodies was performed using the Super Signal West Pico (Thermo Scientific, Hudson, NH, USA) enhanced chemiluminescence substrate and detected with LAS-3000 Plus (Fuji Photo Film, Tokyo, Japan). Relative band intensities were quantified using ImageJ software version 1.53s (National Institute of Health, Bethesda, MD, USA).

### 4.10. RNA Isolation and Quantitative Real-Time PCR Analysis

Total RNA was prepared from cell lines or tissues using TRIzol reagent, according to the manufacturer’s instructions. Then, 5 μg of total RNA was converted into single-stranded cDNA using TOPscript^TM^ cDNA synthesis kit (Enzynomics, Daejeon, Korea). For quantitative real-time PCR analysis, a one-tenth aliquot of cDNA was subjected to PCR amplification using gene-specific primers; mouse F4/80, 5′-TGT GTC GTG CTG TTC AGA ACC-3′ (sense), 5′-AGG AAT CCC GCA ATG ATG G-3′ (antisense); mouse Lcn-2, 5′-GGG AAA TAT GCA CAG GTA TCC TC-3′ (sense), 5′-CAT GGC GAA CTG GTT GTA GTC-3′ (antisense); mouse Gapdh, 5′-AGG TCG GTG TGA ACG GAT TTG-3′ (sense), 5′-GGG GTC GTT GAT GGC AAC A-3′ (antisense). Real-time PCR was performed using SYBR Green PCR Master Mix (Applied Biosystems, Foster City, CA, USA) with an ABI 7500 instrument (Applied Biosystems, Waltham, MA, USA).

### 4.11. Statistical Analysis

At least three independent experiments were conducted. The results are expressed as the means ± standard deviations (SD). Statistical significance between two groups was determined by an unpaired 2-tailed Student’s *t* test. A Shapiro–Wilk test was conducted to check the normality of the data, and Levene’s test was performed to verify the homogeneity of variances before conducting a one-way analysis of variance (ANOVA). ANOVA followed by Scheffe’s test was used for the analysis of differences within each treated condition. Values of *p* < 0.05 indicated statistical significance. *p* < 0.05 indicated statistical significance.

## Figures and Tables

**Figure 1 ijms-23-08899-f001:**
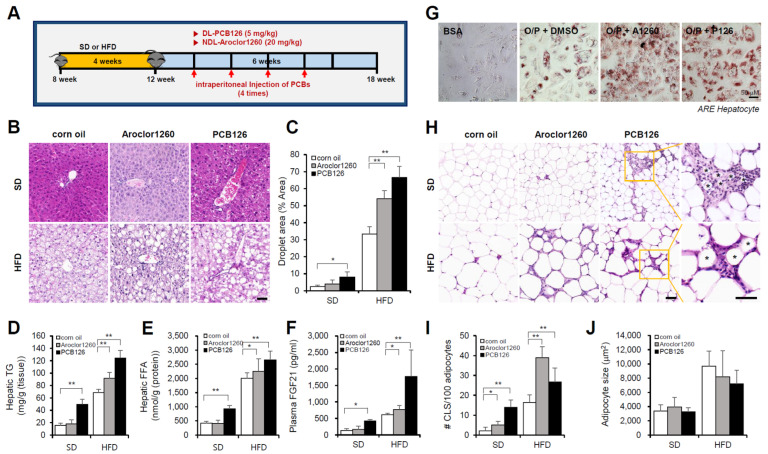
PCB exposure induces NAFLD/NASH in mice fed either a SD and a HFD. Eight-week-old male C57BL/6 mice were fed either a SD or HFD for four weeks and then treated with vehicle (corn oil), Aroclor1260 (20 mg/kg) or PCB126 (5 mg/kg) by intraperitoneal injection for a total of four injections (two, three, four, and five weeks) during the six-week study duration. (**A**) Experimental design. (**B**) Representative images of H&E staining of liver tissue. Scale bar: 50 µm. (**C**) Quantification of the lipid droplet area in H&E-stained liver tissue. The data are expressed as a percentage of the area of lipid droplets in the field. (**D**) Hepatic TG. (**E**) Hepatic FFA. (**F**) Plasma FGF21. (**G**) ORO staining showing that both Aroclor1260 (20 μM) and PCB126 (10 μM) accelerated hepatic lipid accumulation in O/P-treated human ARE primary hepatocytes. Scale bar: 50 µm. (**H**) Representative images of H&E staining of adipose tissue. The crown-like structure is illustrated by the asterisk. Scale bar: 50 µm. (**I**) Average adipocyte size of eWAT was measured in H&E images using ImageJ 1.53 s. (**J**) Quantitative analysis of CLS formation in adipose tissue. All values represent the mean ± SD, n = 7–10 mice per group. * *p* < 0.05 and ** *p* < 0.01.

**Figure 2 ijms-23-08899-f002:**
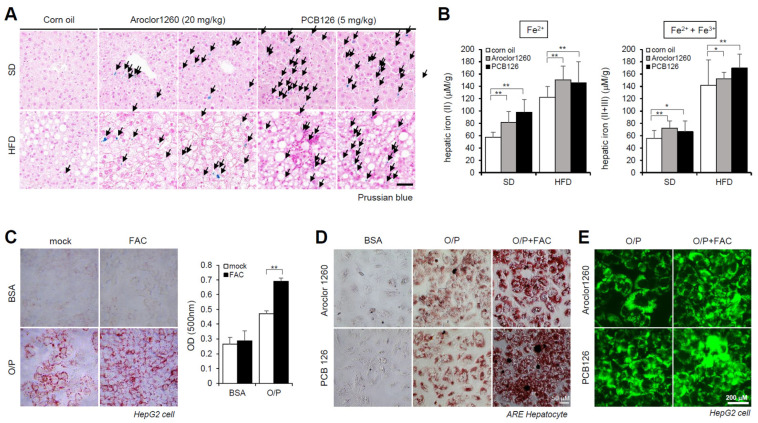
PCB exposure induces hepatic iron overload, and excess iron causes lipid accumulation. (**A**) Prussian blue staining showing marked iron accumulation in liver samples from Aroclor1260- or PCB126 (5 mg/kg)-injected mice fed either a SD or HFD. Scale bar: 50 µm. (**B**) Hepatic iron levels in Aroclor1260- or PCB126-injected mice fed either a SD or HFD (left panel, ferrous ion level; right panel, ferrous and ferric iron total levels). All values represent the mean ± SD, n = six to nine mice per group. * *p* < 0.05 and ** *p* < 0.01. (**C**) ORO staining and quantification showing that 200 μM FAC treatment augments O/P-induced lipid accumulation in HepG2 cells. All values represent the mean ± SD, n = 3. **, *p* < 0.01 compared with the experimental control. (**D**) ORO staining showing that 200 μM FAC accelerated the hepatic lipid accumulation induced by both 20 μM Aroclor1260 and 10 μM PCB126 in O/P-treated human ARE primary hepatocytes. Scale bar: 50 µm. (**E**) BODIPY 493/503 staining showing that 200 μM FAC accelerated the hepatic lipid accumulation induced by both 20 μM Aroclor1260 and 10 μM PCB126 in O/P-treated HepG2 cells. Scale bar: 200 µm.

**Figure 3 ijms-23-08899-f003:**
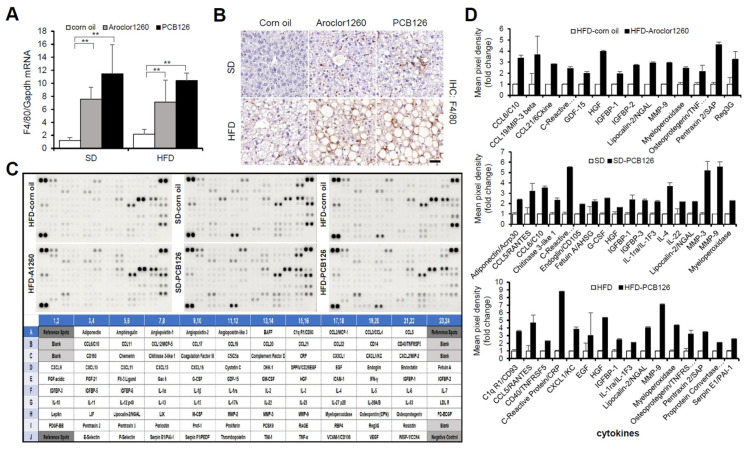
PCB exposure accelerates hepatic inflammation in PCB-induced NAFLD/NASH models. (**A**) Real-time PCR and (**B**) immunohistochemical staining were performed to detect mouse F4/80 in Aroclor1260- or PCB126-injected mice fed either a SD or HFD. Scale bar: 50 µm. All values represent the mean ± SD, n = seven to nine mice per group. ** *p* < 0.01. (**C**) Representative images of cytokine array blots (upper panel) and the alignment of 111 cytokines in duplicates on a mouse XL cytokine array (lower panel). Each blot represents immunoreactive labeling against the respective antibodies. (**D**) Quantification of a mouse XL cytokine array showing that cytokines were significantly increased by Aroclor1260 or PCB126 in mice fed either a SD or HFD. n = 3.

**Figure 4 ijms-23-08899-f004:**
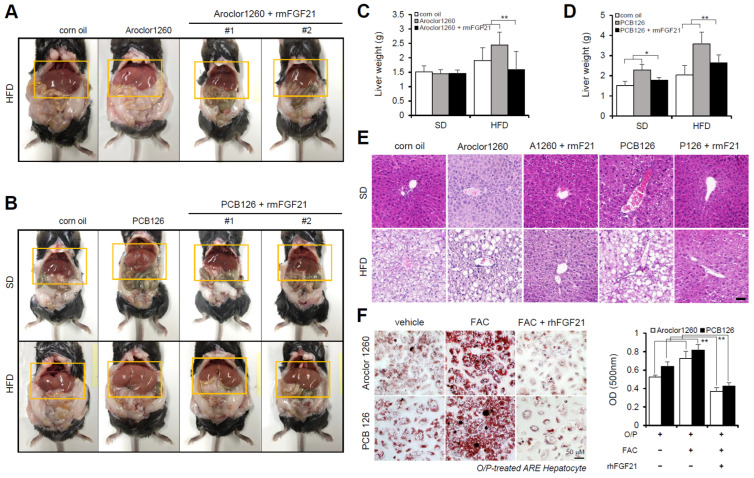
Recombinant FGF21 improves hepatic steatosis in PCB-induced NAFLD/NASH models. Aroclor1260- or PCB126-injected mice fed either a SD or HFD were intraperitoneally administered vehicle or rmFGF21 (1 mg/kg/day) once daily for 10 days. (**A**,**B**) Representative photograph images. (**C**,**D**) Liver weight. (**E**) H,E staining of liver tissue. Scale bar: 50 µm. (**F**) ORO staining and quantification showing that rhFGF21 (50 ng/mL, 24 h) reduced the hepatic lipid accumulation induced by both 20 μM Aroclor1260 and 10 μM PCB126 in O/P-treated human ARE primary hepatocytes. Scale bar: 50 µm. All values represent the mean ± SD, n = 4. * *p* < 0.05, ** *p* < 0.01 compared with the experimental control.

**Figure 5 ijms-23-08899-f005:**
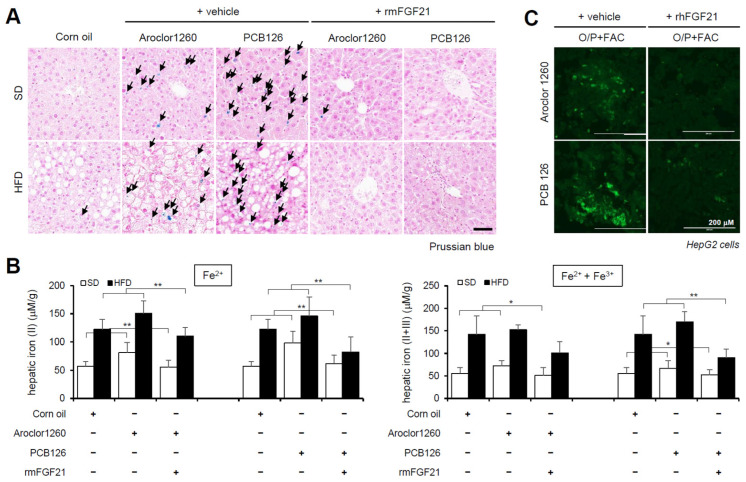
Recombinant FGF21 attenuates hepatic iron overload in PCB-induced NAFLD/NASH models. Aroclor1260 (20 mg/kg)- or PCB126 (5 mg/kg)-injected mice fed either a SD or HFD were intraperitoneally administered vehicle or rmFGF21 (1 mg/kg/day) once daily for 10 days. (**A**) Prussian blue staining showing marked iron accumulation in liver samples. Scale bar: 50 μm. (**B**) Hepatic iron levels in PCB-induced NAFLD/NASH mice that were intraperitoneally administered vehicle or rmFGF21 (left panel, ferrous ion level; right panel, ferrous and ferric iron total levels). All values represent the mean ± SD, n = 7–10 mice per group. * *p* < 0.05 and ** *p* < 0.01. (**C**) Detection of intracellular ferrous ions (Fe^2+^) using the fluorescent probe FerroFarRed in O/P- and FAC-cotreated HepG2 cells. Scale bar: 200 μm.

**Figure 6 ijms-23-08899-f006:**
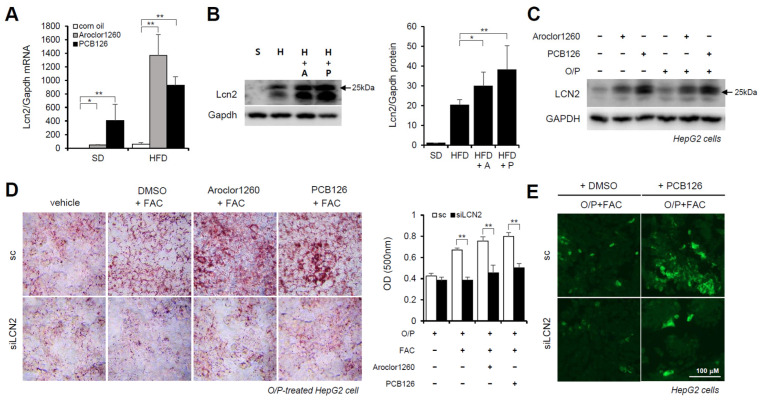
Hepatic LCN2 mediates PCB-induced hepatic lipid and iron accumulation. (**A**) Real-time PCR was performed to detect mouse LCN2 in liver tissues from Aroclor1260- or PCB126-injected mice fed either a SD or HFD. The values represent the mean ± SD, n = six to eight mice per group. * *p* < 0.05 and ** *p* < 0.01. (**B**) Western blot analysis was performed to quantify and detect mouse LCN2 in liver tissues from Aroclor1260- or PCB126-injected mice fed a HFD. (**C**) Western blot analysis was performed to assess human LCN2 in HepG2 cells cotreated with 20 μM Aroclor1260 or 10 μM PCB126 and treated with either BSA or O/P (250 μM). (**D**) ORO staining and quantification showing that silencing human LCN2 attenuated PCB-induced hepatic lipid accumulation in HepG2 cells. All values represent the mean ± SD, n = 4. **, *p* < 0.01 compared with the experimental control. (**E**) Detection of intracellular ferrous ions (Fe^2+^) using the fluorescent probe FerroFarRed showing that silencing human LCN2 reduced PCB126-induced hepatic iron accumulation in O/P-treated HepG2 cells. Scale bar: 100 μm. S, SD; H, HFD; A, Aroclor1260; P, PCB126.

**Figure 7 ijms-23-08899-f007:**
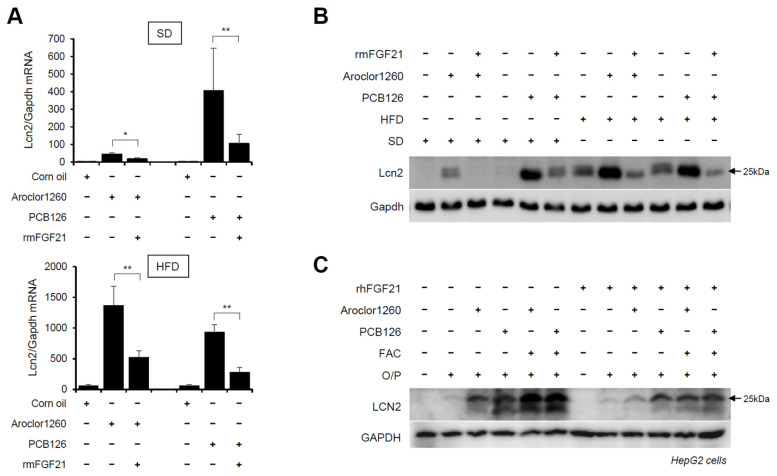
Recombinant FGF21 reduces PCB-induced overexpression of hepatic LCN2. Ten days after the rmFGF21 administration (1 mg/kg/day) in Aroclor1260- or PCB126-induced NAFLD/NASH mice, (**A**) real-time PCR and (**B**) western blot analysis were performed to detect mouse LCN2 in liver tissues from mice fed either a SD or HFD. The values represent the mean ± SD, n = six to eight mice per group. * *p* < 0.05 and ** *p* < 0.01. (**C**) Western blot analysis was performed to assess human LCN2 in 20 μM Aroclor1260- or 10 μM PCB126-treated HepG2 cells cotreated with either vehicle or rhFGF21 (50 ng/mL, 24 h).

## Data Availability

Not applicable.

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
