# Peer review of "Recombinant FGF21 Attenuates Polychlorinated Biphenyl-Induced NAFLD/NASH by Modulating Hepatic Lipocalin-2 Expression"

_ijms, 2022, doi:10.3390/ijms23168899_

Round 1

Reviewer 1 Report

Most of the images are not clear.

For instance Fig.3C: Representative images of cytokine array blots (upper panel) and the alignment of 111 cytokines in duplicates on a mouse XL 189 cytokine array (lower panel), is not clear.

Fig. 6D is not clear.

Fig.7B:  Western blot analysis was performed to assess human LCN2 in 20 265 μM Aroclor1260- or 10 μM PCB126-treated HepG2 cells cotreated with either vehicle or rhFGF21 , western blot is not very clear. needs to be repeated.

Reviewer 2 Report

Dear Authors,

I am really happy to review your recent valuable manuscript, ijms-1821103 which is entitled with 'Recombinant FGF21 attenuates polychlorinated biphenyl-induced NAFLD/NASH by modulating hepatic lipocalin-2 expression'.

Authors well designed and completed their experiments according to their hypothesis; however, this manuscript is needed to revised  prior to accept.

Please refer reviewer's comments as belows;

1. Abstract- Please avoid use the abbreviation without full name on the Abstract section (PCB, NASH, NAFLD)

2. Line 30, please use the full name of NASH.

3. Make sure what is different pathological status between NAFLD and NASH. Please provide more detail on the  'Introduction' section.

4. Please provide the information about HFD diet. What ingredient can be contained? 

5. Four weeks fed of HFD fully induced to accomplish NASH condition? What abut serum liver enzyme levels?
